# Seroprevalence of *Leptospira* spp. Infection in Cattle from Central and Northern Madagascar

**DOI:** 10.3390/ijerph16112014

**Published:** 2019-06-06

**Authors:** Theresa Schafbauer, Anou Dreyfus, Benedikt Hogan, Raphael Rakotozandrindrainy, Sven Poppert, Reinhard K. Straubinger

**Affiliations:** 1Bacteriology and Mycology, Institute for Infectious Diseases and Zoonoses, Department of Veterinary Sciences, Faculty of Veterinary Medicine, LMU Munich, Veterinaerstr. 13, 80539 Munich, Germany; t.schafbauer@lmu.de; 2Swiss Tropical and Public Health Institute, 4051 Basel, Switzerland; anou.dreyfus@swisstph.ch (A.D.); sven.poppert@swisstph.ch (S.P.); 3Faculty of Medicine, University Basel, 4056 Basel, Switzerland; 4Bernhard Nocht Institute for Tropical Medicine, 20359 Hamburg, Germany; hogan@bnitm.de; 5Department of Microbiology and Parasitology, University of Antananarivo, BP 566, Antananarivo 101, Madagascar; rakrapha13@gmail.com

**Keywords:** leptospirosis, Madagascar, cattle, seroprevalence, microscopic agglutination test

## Abstract

Leptospirosis is a zoonotic disease of global importance, especially in tropical countries. The current *Leptospira* spp. seroprevalence in cattle from central and northern Madagascar is unknown. Thus, the aim of this study was to determine the seroprevalence resulting from infections with pathogenic *Leptospira* spp. in zebu cattle from these areas. Serum samples from 194 animals were tested by microscopic agglutination test (MAT) using a panel of 12 serovars as antigens. Samples with a titer of ≥1:100 were considered positive. The overall seroprevalence was 59.3% (95% CI; 52.0–66.2%) with titers ranging from 1:100 to 1:1600. Among the seropositive animals, the most frequent antibody reactions were against serovar *L.* Tarassovi (serogroup *L.* Tarassovi) with 40.2% (33.3–47.5%), followed by *L.* Hardjo (*L.* Sejroe) with 13.9% (9.5–19.8%), *L.* Grippotyphosa (*L.* Grippotyphosa) with 9.8% (6.2–15.1%), *L.* Pomona (*L.* Pomona) with 7.7% (4.5–12.7%) and *L.* Autumnalis (*L.* Autumnalis) with 5.2% (2.6–9.5%). Less than 5% of the samples reacted positively against the remaining serovars. These results indicate a very high exposure of Malagasy cattle to *Leptospira* spp. which, consequently, poses a definite risk for people working with cattle acquiring this zoonotic infection.

## 1. Introduction

Leptospirosis is an important—but neglected—bacterial zoonosis that occurs worldwide, especially in tropical regions. Transmission of *Leptospira* spp. is possible either through direct contact with infected carrier animals or indirectly through contaminated water sources. The most important source of human infection is the brown rat (*Rattus norvegicus*), but many other wild and domestic animals can be reservoir hosts and shed leptospires [1]. Usually the infection source is urine from infected animals, but transmission from cattle to humans is also possible through aborted fetuses or vaginal discharges after abortion or calving [2].

Leptospires are spirochaetes belonging to the genus *Leptospira*, which is divided into 20 genetically different genospecies. In addition, there are presently 24 serogroups with nearly 300 serovars described for pathogenic leptospires [3,4].

Annual cases of human leptospirosis are estimated at 1 million, with 58,900 deaths worldwide, and the majority of cases occur in the tropical regions. Seventy-three percent of cases and deaths are found within the tropical climatic zone between the northern and southern tropics, which includes Madagascar. This makes the disease one of the leading zoonotic causes of morbidity and mortality in the human population [5].

In Madagascar, studies about leptospirosis are rare and, for a while, it was not obvious if *Leptospira* spp. were present on the island or not. There are a few studies pertaining to leptospiral infection in humans that have been carried out so far. The earliest evidence of human leptospirosis was reported in 1956 [6], followed by additional described cases in 1968, 1978, and 2001 [7,8,9]. All cases were confirmed using a microscopic agglutination test (MAT), which is the gold standard in serological diagnosis. The results of the MAT showed antibody reactions against the following serogroups: *L.* Australis, *L.* Canicola, *L.* Grippotyphosa, *L.* Hebdomadis, *L.* Icterohaemorrhagiae and *L.* Tarassovi. Cattle are incidental hosts for all of these serogroups [2]. In 2015, leptospire organisms were detected and confirmed in a human case using molecular methods, specifically the polymerase chain reaction (PCR), for the first time [10]. A cross-sectional study on human leptospirosis in Madagascar from 2015 found a seroprevalence of 2.9% (*n* = 678) in healthy and febrile humans; cattle were a significant risk factor for seropositivity (odds ratio [OR] = 3.4, *p*-value = 0.02) [11]. Based on this risk-factor analysis, a cattle to human transmission pathway can be hypothesized.

Infection by *Leptospira* spp. can cause severe disease in cattle, such as pyrexia, hemolytic anemia, jaundice, meningitis, or even death, but these cases are uncommon and very rare. The infection with serovar *L.* Hardjo, for which cattle are known to be a maintenance host, is often subclinical. Importantly, however, the economic losses for farmers resulting from reproductive disorders or abortion are not negligible, and affected cattle can pose a possible threat to the health of their owners. Moreover, infection in dairy cows may sometimes show as an acute milk drop syndrome [2].

Data about leptospirosis in Malagasy cattle is very scarce. The only reports on MAT-confirmed leptospirosis in sick and healthy animals from eastern and central Madagascar were published in 1956 [6,12]. In the southern part of the island, a high seroprevalence in healthy cattle of more than 36% was found using MAT in 1968 [7]. Information about leptospirosis in cattle from northern Madagascar is not available. The most recent study tried to detect DNA from *Leptospira* spp. in bovine kidney samples with PCR, but without positive results [9]. However, it should be noted that the number of tested animals was very small and no MAT was performed. Thus, it was not clear whether the tested animals had any contact with leptospires.

The present study aimed to obtain more information about the current infection status of cattle from central and northern Madagascar with *Leptospira* spp. and, in the case of high seropositivity, reveal a possible public health risk.

## 2. Materials and Methods

### 2.1. Animals and Sample Collection

Serum samples from 201 zebu cattle were collected in three different slaughterhouses in the municipality of Bemasoandro (district of Antananarivo-Atsimondrano) in Madagascar in October 2012. All blood samples were left to clot at ambient temperature before the serum was collected and stored at −20 °C. Samples of these animals were investigated prior to the present work in the scope of a study on different zoonotic diseases [13,14,15].

Afterwards, the samples were delivered on dry ice to the Institute of Infectious Diseases and Zoonoses, LMU Munich, Germany for serological analysis and were stored at −80 °C until testing.

### 2.2. Serological Analysis

To detect antibodies against pathogenic *Leptospira* spp., all serum samples were tested with the MAT, in accordance with the methodology of Goris and Hartskeerl [16]. Cultures with living organisms of 12 *Leptospira* spp. reference strains were used in this study (Table 1).

Serovars were chosen based on previously published data from Madagascar [6,7,12]. All reference strains were originally supplied by the Federal Institute of Risk Assessment (Berlin, Germany) except strain Veldrat Batavia 46, which came from the Academic Medical Center (Amsterdam, The Netherlands). Negative (phosphate-buffered saline, PBS) and positive (rabbit antisera) controls were included in the test protocol. Rabbit antisera were also supplied by the Academic Medical Center (Amsterdam, The Netherlands). Leptospiral strains used in the MAT were cultured in liquid Ellinghausen–McCullough–Johnson–Harris (EMJH) medium (Becton, Dickinson and Company, Sparks, MD, USA), incubated at 29 °C, and used after 7 to 11 days of culture. For standardization, leptospires were counted with a Petroff–Hausser counting chamber (Hausser Scientific, Horsham, PA, USA) and adjusted to approximately 2 × 10^8^ organisms per mL.

To protect the laboratory personnel from infection, all serum samples were heat-inactivated for 30 min in a water bath at 56 °C [16].

All samples were initially screened at a dilution of 1:50 with an inverse dark-field microscope (Leica DMi8, Leica Microsystems, Wetzlar, Germany) using 100× total magnifying power. Sera with a positive reaction in this screening were titrated in a serial twofold dilution from 1:50 to 1:6400 to determine the end-point titer. The antibody titer is defined as the highest dilution containing ≥50% of agglutinated leptospires. Samples with a titer of ≥1:100 were considered positive. If no agglutination was detectable, the sample was considered negative.

### 2.3. Statistical Analysis

Data was organized and analyzed with Microsoft Excel 2016 (Microsoft Corporation, Redmond, WA, USA) and OriginPro 9.1 (OriginLab Corporation, Northampton, MA, USA). The overall seroprevalence, prevalence for each serovar, 95% confidence intervals (CI), and geometric mean titer for each serovar were calculated [17]. Differences in the *Leptospira* spp. seroprevalence and seroprevalences of each serovar according to geographic sampling regions were analyzed by a chi-square test. A *p*-value less than 0.05 was considered significant.

## 3. Results

### 3.1. Description of Study Population and Tested Samples

A total of 201 serum samples were collected from cattle in three slaughterhouses located close to Antananarivo, the capital of Madagascar. The cattle originated from five different regions in central and northern Madagascar: Bongolava (*n* = 79; 39.3%), Haute Matsiatra (*n* = 63; 31.3%), Menabe (*n* = 27; 13.4%), Sofia (*n* = 22; 10.9%) and Vakinankaratra (*n* = 10; 5.0%) (Figure 1).

All cattle were pure *Bos indicus* breed and exclusively male with an age between 4 and 20 years. Two animals were classified as sick during routine veterinary examination; all other cattle were considered healthy.

From these 201 collected samples, seven samples could not be evaluated due to insufficient serum quality or small amounts of serum. Finally, 194 samples were tested serologically using MAT.

### 3.2. Serological Results

Of the 194 tested sera, a total of 115 samples had a positive titer of 1:100 or higher against one or more serovars, which results in an overall seroprevalence of 59.3% (95% CI; 52.0–66.2%). The most reactive serovar was *L.* Tarassovi (serogroup *L.* Tarassovi) with 40.2% (33.3–47.5%), followed by *L.* Hardjo (*L.* Sejroe) with 13.9% (9.5–19.8%), *L.* Grippotyphosa (*L.* Grippotyphosa) with 9.8% (6.2–15.1%), *L.* Pomona (*L.* Pomona) with 7.7% (4.5–12.7%) and *L.* Autumnalis (*L.* Autumnalis) with 5.2% (2.6–9.5%). Serovars *L.* Pyrogenes (serogroup *L.* Pyrogenes), *L.* Bataviae (*L.* Bataviae), *L.* Australis (*L.* Australis), *L.* Javanica (*L.* Javanica) and *L.* Ballum (*L.* Ballum) showed a seroprevalence of less than 5% each (Table 2). No sample showed a positive reaction against serovar *L.* Canicola (*L.* Canicola) or *L.* Icterohaemorrhagiae (*L.* Icterohaemorrhagiae).

The *Leptospira* spp. seroprevalence among the five sampling areas showed no statistically significant difference (*p*-value = 0.71). Moreover, there were no statistically significant differences between the seroprevalences of each serovar among the five different sampling regions (Appendix A).

Reactions with two or more serovars were detected in 42/115 samples (36.5%). Most reactions against more than one serovar in a single serum sample were observed for *L.* Tarassovi with *L.* Hardjo (Table 3).

MAT titers ranged from 1:100 to 1:1600, with one sample having a positive titer of 1:1600 against serovar *L.* Grippotyphosa (Figure 2, Appendix A). The geometric mean titer ranged from 1:100 (*L.* Ballum) to 1:247 (*L.* Grippotyphosa) (Figure 3).

## 4. Discussion

The aim of this study was to evaluate the seroprevalence of *Leptospira* spp. in cattle from central and northern Madagascar, indicating past exposure to leptospires. For all the data presented here, it has to be considered that we only tested male cattle and the situation may be different for female cattle. The obtained seroprevalence of 59.3% shows high exposure of cattle to *Leptospira* spp. and a possible public health risk for people working with cattle. According to the data presented here, leptospirosis appears to be a major infectious disease in Malagasy cattle, which is in agreement with data from Reunion Island (also part of the Western Indian Ocean islands) [18]. In a similar study from 1968 that focused on leptospirosis in cattle from southern Madagascar, the seroprevalence was 36.1% (*n* = 72) [7]. Factors associated with such a high seroprevalence in cattle may be due to contact with other animals, such as domesticated or wild pigs, sheep, rodents or other small mammals, which also carry leptospiral infections and transmit the organisms via urine. Another important source of infection could be the common usage of water sources. The transmission of leptospires via venereal spread is also a factor that should not be underestimated.

Among the five geographic sampling regions, there was no significant statistical difference in seroprevalence. The main reason for this is probably the sample size, which was not large enough to detect small differences. Another factor could be the very similar climatic conditions in these regions. Usually high humidity and rainfall are associated with cases of leptospirosis, because under these conditions most leptospires have a very high survival rate in the environment. Central and northern Madagascar is known for its sub-humid to humid climate, in contrast to the (semi-)arid southern part of the island [19]. The tropical climate on islands in the Western Indian Ocean consists of two main seasons: A hot and rainy season, the austral summer, and a dry season, the austral winter [18]. As our sampling was conducted during the austral summer, the exposure to leptospires may have been slightly higher than during the dry winter period.

The most reactive serovar in 40.2% of the tested samples was *L.* Tarassovi (genospecies *L. borgpetersenii*), which corresponds with data from cattle in southern Madagascar [7]. *L.* Tarassovi is well-known for infections in beef cattle from New Zealand and Australia, with herd seroprevalences of 74.0% and 13.9%, respectively [20,21]. A recent study in Thailand detected a herd-seroprevalence of 92.9% for *L.* Tarassovi [22]. Although, in these studies, a MAT cut-off titer of 1:48 or 1:50 was used, both of which are lower than the one we used in our study. Additionally, pigs and wild boars are known to be maintenance hosts for serovar *L.* Tarassovi. In Madagascar, most pigs are located in the central and northern part of the island; though, data about the population of wild boars are missing [19]. As contact with wild boars or pigs may occur during grazing, their role in the pig–cattle transmission cycle seems to be quite important. Note that, due to cross-reactivity in the MAT, it is possible that another serovar indicated the antibody reaction for *L.* Tarassovi, and therefore its seroprevalence could be overestimated.

Serovar *L.* Hardjo (genospecies *L. interrogans*) is usually associated with cattle, as they are known to be a maintenance host. The observed seroprevalence of 13.9% shows that *L.* Hardjo is present in Malagasy cattle. Currently, information about the genospecies *L. interrogans* and *L. borgpetersenii* on Madagascar is missing. Thus, as these genospecies are likely to cross-react in the MAT, it may be possible that the detected antibodies are against *L. borgpetersenii* serovar Hardjo, but not *L. interrogans* serovar Hardjo. Transmission of serovar *L.* Hardjo is possible through placental infection or direct contact with urine or fetal membranes from infected cattle [2]. The animals in our study were exclusively male, so they were probably already infected as a fetus or during birth. Transmission from other infected cattle may also be possible. In maintenance hosts, infection is often subclinical or the clinical signs are very mild. Chronic leptospirosis, on the other hand, can lead to abortion, stillbirth or infertility in female cattle [2]. The economic impact of the infection in cattle from Madagascar is still unknown.

The obtained seroprevalence of 9.8% for the rodent-associated serovar *L.* Grippotyphosa (genospecies *L. kirschneri*) shows that this classical transmission pathway, perpetuated by small mammals, also plays a role in Malagasy cattle. Madagascar, with its diversity of ecosystems and mammal populations, provides a very wide range of potential hosts for *Leptospira* spp. [18]. Additionally, endemic small mammals (e.g., rodents, tenrecs and shrews) and bats may have a role in the transmission cycle that includes cattle, as they harbor leptospires from the same genospecies like the ones found in our study [23,24].

Exposure to two or more serovars, or exposure to one serovar with cross-agglutinations, was found in 36.5% of the tested samples. Cross-reactions may occur, since several common antigens are found among leptospires. During the acute phase of an infection, IgM antibodies may be present at elevated levels, followed later by the production of more specific IgG antibodies. IgM may also contribute to a certain degree of antibody cross-reaction. After an infection has been overcome, antibody titers can decline at very different rates. Especially following a case of acute infection, some titers may remain high for a long time and may take months or even years to decline to lower levels. Thus, if a sample shows cross-reactions, the highest titer does not necessarily indicate the infecting serovar. However, as shown previously, it does indicate that the animal was at least, at one time, infected by a strain belonging to this serogroup [25]. Infecting serovars can only be identified with isolates from infected tissues or fluids, which is rather difficult, or with PCR, which can be run directly from urine or tissues. The research community is divided regarding the interpretation of MAT results. Some argue that data obtained with MAT only provide a general impression about the presence of serogroups within a population or within the environment [26]. However, if the epidemiology of leptospirosis is well known, and the MAT validated in the context of a country, MAT results can be serovar-specific [27]. Given the unknown epidemiology in Madagascar, MAT interpretation regarding serovars should be made with caution.

Antibody titers in our study ranged from 1:100 to 1:1600. According to the OIE (Office International des Epizooties, World Organization for Animal Health) a titer of ≥1:100 is considered positive [28]. Yet, clinically relevant titers in cases of acute leptospirosis are usually much higher [26]. In our study, all animals were considered healthy during routine veterinary examination with two exceptions, from which one showed an antibody titer against serovar *L.* Tarassovi of 1:200. As the titer of this case is reasonably low, acute leptospirosis is not expected. 

Most of the antibody levels we detected were low (≤1:400). They probably resulted from earlier infections with antibodies remaining in the sera or very early infections without clinical signs. High titers of 1:800 or 1:1600 were found for serovars *L.* Grippotyphosa, *L.* Hardjo, *L.* Pomona, *L.* Pyrogenes and *L.* Tarassovi. The affected animals could have had acute leptospirosis without obvious clinical signs, or the titers may have resulted from older infections with slowly decreasing (but remaining) antibody levels. Another reason could be a reinfection, wherein antibodies from an older infection were still present, and thus the titer was boosted because of the production of new antibodies. High antibody titers are also possible during the early weeks after vaccines have been applied. As this latter scenario is not practiced in Madagascar, vaccination can be excluded as a possible reason for the antibody positivity.

With 194 samples, the sample size was large enough to precisely estimate an apparent prevalence of 15%, with a precision of 0.05 and confidence of 95%. Hence, our sample size was too small to get a precise estimate for the existing prevalence, which was reflected in the wider confidence intervals. A limitation of the study is the small and varying number of samples from each geographic region. Further, as the samples were collected in slaughterhouses, the origin of the animals was unknown and equal distribution of the areas was not possible. Therefore, we did not calculate the seroprevalence for each sampling region, but for the central and northern part of Madagascar in total.

From our study, we can show that cattle from central and northern Madagascar have been exposed to *Leptospira* spp., but the number of cattle suffering from leptospirosis with clinical signs is still unknown. Moreover, the importance of leptospirosis in humans in Madagascar is not clear, as there are only a few reports; for example, a recent study in fever patients (*n* = 1009) did not detect leptospiral DNA with real-time PCR [15]. In particular, areas with rice cultivation, where cattle and rodents occur, infection in humans can be suspected.

To evaluate the public health relevance, the detection of leptospiral DNA in bovine kidney or urine samples and the status of cattle as shedders of pathogenic leptospires is highly recommended as the next steps in future research. Additionally, insight on the seroprevalence in female cattle and the economic impact of *Leptospira* spp. infection on Malagasy cattle may also assist in assessing the importance of future leptospirosis control in Madagascar. Finally, further studies on leptospirosis in humans are highly recommended to evaluate the current infection status of this widespread zoonotic disease.

## 5. Conclusions

A seroprevalence of 59.3% is strong evidence for exposure of cattle from central and northern Madagascar to pathogenic leptospires. Hence, transmission of *Leptospira* spp. from livestock to humans is possible and should be considered as a risk factor relevant for public health aspects. Based on these findings, we recommend further studies focusing on detection of leptospire organisms in Malagasy cattle, rodents and humans to reveal and characterize the shedding status of affected animals and herds, and to determine the relevance for humans. Beyond the detection of leptospires, it is also advised to search for possible causes of Leptospirosis on Madagascar. Moreover, humans on Madagascar should be informed adequately about this zoonotic disease and how they can prevent themselves from being infected. For example, by avoiding contact with rodents or the urine of cattle, or the use of protective gloves during work with cattle.

## Figures and Tables

**Figure 1 ijerph-16-02014-f001:**
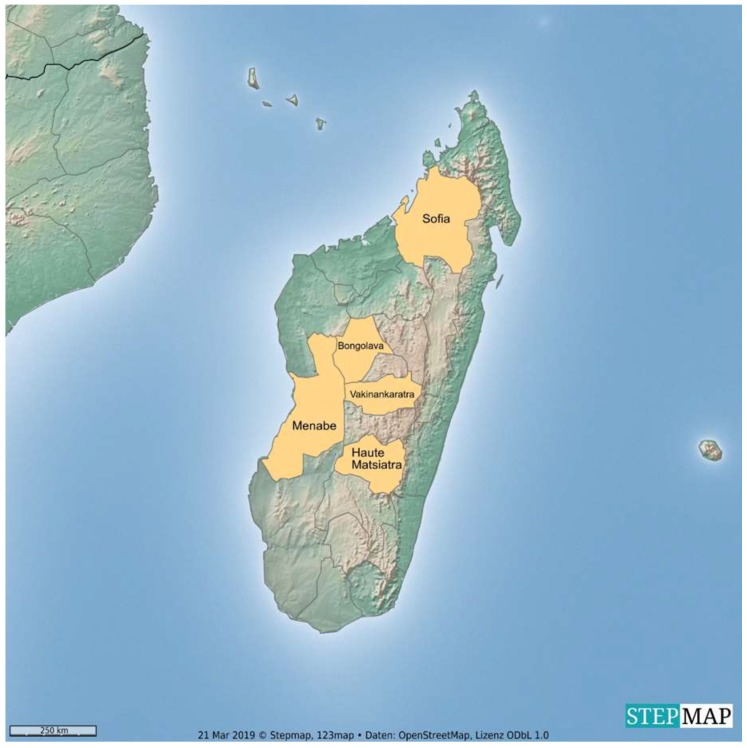
Origin of Malagasy zebu sera.

**Figure 2 ijerph-16-02014-f002:**
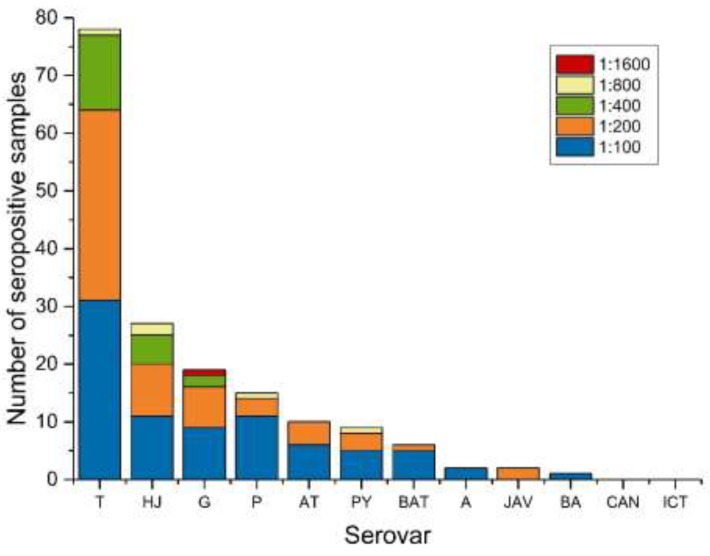
Number of seropositive samples with titers from 1:100 to 1:1600 for each serovar. Seroprevalences of some serovars may be overestimated due to cross-reactivity. Serovars included in the test: *L.* Tarassovi (T), *L.* Hardjo (HJ), *L.* Grippotyphosa (G), *L.* Pomona (P), *L.* Autumnalis (AT), *L.* Pyrogenes (PY), *L.* Bataviae (BAT), *L.* Australis (A), *L.* Javanica (JAV), *L.* Ballum (BA), *L.* Canicola (CAN) and *L.* Icterohaemorrhagiae (ICT).

**Figure 3 ijerph-16-02014-f003:**
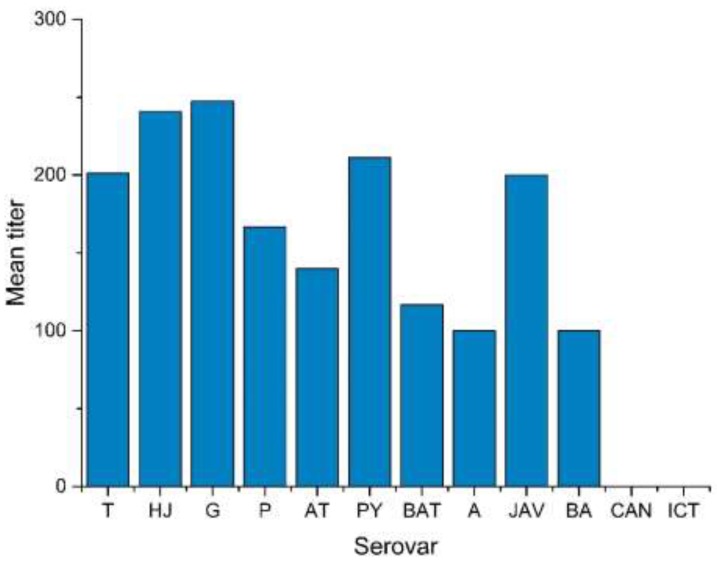
Mean titer of each serovar. Serovars included in the test: *L.* Tarassovi (T), *L.* Hardjo (HJ), *L.* Grippotyphosa (G), *L.* Pomona (P), *L.* Autumnalis (AT), *L.* Pyrogenes (PY), *L.* Bataviae (BAT), *L.* Australis (A), *L.* Javanica (JAV), *L.* Ballum (BA), *L.* Canicola (CAN) and *L.* Icterohaemorrhagiae (ICT)

**Table 1 ijerph-16-02014-t001:** Panel of *Leptospira* spp. strains used as antigens in the microscopic agglutination test (MAT).

Genospecies	Serogroup	Serovar	Strain
*L. interrogans*	Australis	Australis	Ballico
	Autumnalis	Autumnalis	Akiyami A
	Bataviae	Bataviae	Swart
	Canicola	Canicola	Hond Utrecht IV
	Icterohaemorrhagiae	Icterohaemorrhagiae	Ictero I
	Pomona	Pomona	Pomona
	Pyrogenes	Pyrogenes	Salinem
	Sejroe	Hardjo	Hardjoprajitno
*L. borgpetersenii*	Ballum	Ballum	Mus 127
	Javanica	Javanica	Veldrat Batavia 46
	Tarassovi	Tarassovi	Perepelitsin
*L. kirschneri*	Grippotyphosa	Grippotyphosa	Moskva V

**Table 2 ijerph-16-02014-t002:** Number of seropositive samples and seroprevalence of *Leptospira* spp. serovars using MAT (titer ≥ 1:100); in total 194 bovine serum samples from central and northern Madagascar were included in the study.

Serovar	Seropositive Samples *n*	Seroprevalence %	CI 95%
*L.* Tarassovi	78	40.2	33.3–47.5
*L.* Hardjo	27	13.9	9.5–19.8
*L.* Grippotyphosa	19	9.8	6.2–15.1
*L.* Pomona	15	7.7	4.5–12.7
*L.* Autumnalis	10	5.2	2.6–9.5
*L.* Pyrogenes	9	4.6	2.3–8.9
*L.* Bataviae	6	3.1	1.3–6.9
*L.* Australis	2	1	0.2–4.1
*L.* Javanica	2	1	0.2–4.1
*L.* Ballum	1	0.5	0.0–3.3
*L.* Canicola	0	0	-
*L.* Icterohaemorrhagiae	0	0	-

**Table 3 ijerph-16-02014-t003:** Number and percentage of positive samples reacting with two, three or more serovars.

Positive Serovars *n*	Samples with Multiple Reactions *n* (%)	Serovars with Frequent Reactions (*n*)
2	32 (27.8)	T ^1^ + HJ ^2^ (9)
3	8 (7.0)	
4	2 (1.7)	

^1^*L.* Tarassovi, ^2^
*L.* Hardjo.

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
