# Peer review of "Seroprevalence of Leptospira spp. Infection in Cattle from Central and Northern Madagascar"

_ijerph, 2019, doi:10.3390/ijerph16112014_

Round 1
Reviewer 1 Report
THIS MANUSCRIPT SHOWS RESULTS PROVIDED BY A LOW NUMBER OF SAMPLES BUT IT IS VERY INTERESTING. HOWEVER, TO BECOME PERFECT SOME DETAILS MISSING IN THE DISCUSSION HAVE TO BE ADDED OR MODIFIED.
63The infection with serovar L. Hardjo, for which cattle are known to be a maintenance host, is often sublinical. OK, BY TAKING INTO ACCOUNT COW BY COW, BUT ECONOMIC LOSS FOR THE FARMERS ARE DUE TO REPRODUCTIVE FAILURE AND ABORTION ASIDE THEIR OWN HEALTH RISK;
124 exclusively male with an age between four and 20 years.IT IS A HEAVY STATISTICAL BIAS WHICH HAS TO BE HIGHLY DISCUSSED! ESPECIALLY TAKING INTO ACCOUNT HJ RESULTS 186Transmission of serovar L. Hardjo is possible through placental infection or direct contact with urine or fetal membranes from infected cattle [2]. In maintenance hosts, infection is often 187subclinical or the clinical signs are very mild.
196Exposure to two or more serovars was found in 36.5% of the tested samples. OR EXPOSURE TO ONE SEROVAR WITH CROSS-AGGLUTINATIONS. IT WOULD BE INTERESTING TO CHECK THE GEOMETRIC MEAN TITER FOR EACH SEROGROUP INSTEAD OF THE FIG 2; THIS FIG IS POORLY INFORMATIVE FOR READERS. THEY COULD UNDERSTAND THAT TARASSOVI IS THE MOST INFECTIOUS. MAY BE NOT, BECAUSE AS SHOWN MOST OF TARASSOVI TITERS ARE LOW AND COULD THEREFORE BE CONSIDERED AS CROSS-AGGLUTINATIONS.
144 Most reactions against more than one serovar in a single serum sample were observed for L. Tarassovi with L. Hardjo. OK, SEE ATTACH FILE (USING THE NUMBERS FOUND IN THE SUPPLEMENTARY TABLE) SHOWING THAT WHEN A SERUM IS POSITIVE FOR TARASSOVI,, THE TITER IS GENERALLY LOWER THAN FOR HJ AND EVEN GRIP…
201Especially after a case of acute infection, some titers may remain high for a long time and may take months 202 or even years to decline to lower levels. CATTLE ARE HIGHLY EXPOSED TO LEPTO INFECTION. THUS THE IMMUNE SYSTEM OF AN ANIMAL WHICH WAS PREVIOUSLY PRINTED BY ALL THE ANTIGENS SHARED BY THE PREVIOUS INFECTIOUS LEPTO SPP (EVEN IF ANOTHER SEROVAR) IS BOOSTED BY NEW (SELF-LIMITED) INFECTION. “ The affected animals could have had an acute leptospirosis without obvious clinical signs or the titers resulted from older infections with slowly decreasing, remaining 225 antibody levels. 226
THEREFORE” Thus, if a sample shows cross-reactions, the highest titer not necessarily indicates the infecting serovar.” BUT STATISTICALLY IT INDICATES THAT THE ANIMAL WAS AT LEAST ONCE INFECTED BY A STRAIN BELONGING TO THIS SEROGROUP AS DEMONSTRATED BY R. MOROZETTI BLANCOA : IS
THE MICROAGGLUTINATION TEST (MAT) GOOD FOR PREDICTING THE INFECTING SEROGROUP FOR LEPTOSPIROSIS IN BRAZIL? COMP. IMMUNOL. MICROBIOL. INFECT. DIS. 44 (2016) 34-36.
However, if the epidemiology of leptospirosis is well known and the MAT validated in the context of a country, 208MAT results can be serovar specific [26]. Given the unknown epidemiology in Madagascar, MAT 209interpretation regarding serovars should be made with caution. .OK.IT’S FOR THIS REASON THAT INFECTIOUS SEROGROUPS ONLY CAN BE IDENTIFIED AND FOR THIS REASON TOO, IT IS GENERALLY BETTER TO USE MORE THAN ONE SEROVAR FOR A GIVEN SEROGROUP, ESPECIALLY ICTEROHAEMORRHAGIAE, THE MOST PROMINENT IN HUMANS. THE UNIC STRAIN USED IN THIS STUDY IS WELL KNOWN TO GIVE POOR AGGLUTINATING LEVEL, COMPARED TO OTHER ICTERO STRAINS AND TO THE CLOSE SEROVAR COPENHAGENI.:137No sample showed a positive reaction against serovar L. Canicola (L. Canicola) or L. Icterohaemorrhagiae (L. Icterohaemorrhagiae).
214 To verify the diagnosis “leptospirosis” with certainty, a paired serum sample collected three weeks apart is 215 required. A fourfold or greater rise in antibody titers between the two sera confirms the diagnosis [25].THIS MAT INTERPRETATION IS TRUE FOR ACUTE CASES OF LEPTOSPIROSIS AS DIAGNOSED IN HUMANS AND CANINE WHICH EXHIBIT INDIVIDUAL DISEASE BUT NOT IN CATTLE. LEPTOSPIROSIS IN CATTLE IS NOT A DISEASE OF ONE ANIMAL. IT’S AN INFECTION SPREADING IN THE WHOLE HERD INDUCING SELDOM ACUTE LEPTOSPIROSIS EXPRESSED AS THE MILKING DROP SYNDROM, BUT MOSTLY ABORTIONS AND REPRODUCTIVE FAILURE WHICH ARE NOT ACUTE. THEREFORE HIGH TITERS CAN BE SHOWN BY THE NEIGHBOUR OF THE ABORTED COW WHEREAS THE TITERS OF THE ABORTED COW ITSELF YET DROPPED DOWN, BECAUSE AS INDICATED: 63The infection with serovar L. Hardjo, for which cattle are known to be a maintenance host, is often sublinical. .. 186Transmission of serovar L. Hardjo is possible through placental infection or direct contact with urine or fetal membranes from infected cattle [2]. In maintenance hosts, infection is often 187subclinical or the clinical signs are very mild.
262 We also thank Ivonne Stamm from IDEXX Laboratories, Ludwigsburg, Germany for her technical advice concerning the MAT.
WHY IDEXX?? THIS COMPANY IS NOT SPECIALIZED IN MAT, BUT IN OTHER METHODS AS ELISA....IT WOULD HAVE BE MORE JUSTIFIED TO DISCUSS WITH ACADEMIC OR REFERENCE VET CENTERS.ONSET OF HUMAN LEPTOSPIROSIS IS STRONGLY DIFFERENT FROM HERD LEPTOSPIROSIS? EVEN IF IMMUNOLOGICAL EVENTS ARE SIMILAR;
Author Response
Cover letter Munich, June 3rd, 2019
Dear Editors, dear Referees,
thank you for the quick processing of our manuscript “Seroprevalence of Leptospira spp. Infections in Cattle from Central and Northern Madagascar” and for your constructive comments. We provide our response and explanations to your comments in red fonts.
Concerning the report of the reviewer #1:
1. 63 The infection with serovar L. Hardjo, for which cattle are known to be a maintenance host, is often sublinical. OK, BY TAKING INTO ACCOUNT COW BY COW, BUT ECONOMIC LOSS FOR THE FARMERS ARE DUE TO REPRODUCTIVE FAILURE AND ABORTION ASIDE THEIR OWN HEALTH RISK;
We added missing information regarding reproductive disorders and the owners' health risk (lines 64-66).
“Even though the economic losses for farmers due to reproductive disorders or abortion are not negligible. Moreover, affected cattle can pose a possible threat to the health of their owners.”
2. 124 exclusively male with an age between four and 20 years.IT IS A HEAVY STATISTICAL BIAS WHICH HAS TO BE HIGHLY DISCUSSED! ESPECIALLY TAKING INTO ACCOUNT HJ RESULTS 186Transmission of serovar L. Hardjo is possible through placental infection or direct contact with urine or fetal membranes from infected cattle [2]. In maintenance hosts, infection is often 187subclinical or the clinical signs are very mild.
In the discussion we included the factor that we have only tested male cattle (lines 172-174 and lines 212-214).
“For all the data presented here, it has to be considered that we only tested male cattle and the situation may be different for female cattle.”
“The animals in our study were exclusively male so they were probably already infected as a fetus or during birth. Transmission from other infected male cattle may also be possible.”
Moreover, we included the need of a seroprevalence study in female cattle into the future research paragraph (line 275).
insight on “the seroprevalence in female cattle”
3. 196 Exposure to two or more serovars was found in 36.5% of the tested samples. OR EXPOSURE TO ONE SEROVAR WITH CROSS-AGGLUTINATIONS. IT WOULD BE INTERESTING TO CHECK THE GEOMETRIC MEAN TITER FOR EACH SEROGROUP INSTEAD OF THE FIG 2; THIS FIG IS POORLY INFORMATIVE FOR READERS. THEY COULD UNDERSTAND THAT TARASSOVI IS THE MOST INFECTIOUS. MAY BE NOT, BECAUSE AS SHOWN MOST OF TARASSOVI TITERS ARE LOW AND COULD THEREFORE BE CONSIDERED AS CROSS-AGGLUTINATIONS.
We inserted “or exposure to one serovar with cross-agglutinations” in line 223.
We calculated the mean titer for each serovar (line 116, lines 155/156) and included the corresponding figure into the text (Figure 3).
Following insertions were added (line 154; lines 164-167).
“The geometric mean titer ranged from 1:100 (L. Ballum) to 1:247 (L. Grippotyphosa) (Figure 3).”
“Figure 3. Mean titer of each serovar. Serovars included in the test: L. Tarassovi (T), L. Hardjo (HJ), L. Grippotyphosa (G), L. Pomona (P), L. Autumnalis (AT), L. Pyrogenes (PY), L. Bataviae (BAT), L. Australis (A), L. Javanica (JAV), L. Ballum (BA), L. Canicola (CAN) and L. Icterohaemorrhagiae (ICT).”
Moreover, we included the following sentence into the legend of Figure 2 to avoid misinterpretation
“Seroprevalences of some serovars may be overestimated due to cross-reactivity.”
4. 144 Most reactions against more than one serovar in a single serum sample were observed for L. Tarassovi with L. Hardjo. OK, SEE ATTACH FILE (USING THE NUMBERS FOUND IN THE SUPPLEMENTARY TABLE) SHOWING THAT WHEN A SERUM IS POSITIVE FOR TARASSOVI,, THE TITER IS GENERALLY LOWER THAN FOR HJ AND EVEN GRIP…
We included this information into the discussion (lines 203-205).
“Due to cross-reactivity in the MAT, it may be possible that another serovar indicated the antibody reaction and the seroprevalence of L. Tarassovi could be overestimated.”
5. 201 Especially after a case of acute infection, some titers may remain high for a long time and may take months 202 or even years to decline to lower levels. CATTLE ARE HIGHLY EXPOSED TO LEPTO INFECTION. THUS THE IMMUNE SYSTEM OF AN ANIMAL WHICH WAS PREVIOUSLY PRINTED BY ALL THE ANTIGENS SHARED BY THE PREVIOUS INFECTIOUS LEPTO SPP (EVEN IF ANOTHER SEROVAR) IS BOOSTED BY NEW (SELF-LIMITED) INFECTION. “ The affected animals could have had an acute leptospirosis without obvious clinical signs or the titers resulted from older infections with slowly decreasing, remaining 225 antibody levels. 226
We included the point regarding the boosted new infection in our discussion (lines 255/256).
“Another reason could be a reinfection where antibodies from an older infection are still present and the titer is boosted because of the production of new antibodies.”
6. THEREFORE” Thus, if a sample shows cross-reactions, the highest titer not necessarily indicates the infecting serovar.” BUT STATISTICALLY IT INDICATES THAT THE ANIMAL WAS AT LEAST ONCE INFECTED BY A STRAIN BELONGING TO THIS SEROGROUP AS DEMONSTRATED BY R. MOROZETTI BLANCOA: IS THE MICROAGGLUTINATION TEST (MAT) GOOD FOR PREDICTING THE INFECTING SEROGROUP FOR LEPTOSPIROSIS IN BRAZIL? COMP. IMMUNOL. MICROBIOL. INFECT. DIS. 44 (2016) 34-36.
We added this information to the discussion and inserted the reference you cited (lines 230-232).
“But as shown previously, it indicates that the animal was at least once infected by a strain belonging to this serogroup.”
7. However, if the epidemiology of leptospirosis is well known and the MAT validated in the context of a country, 208MAT results can be serovar specific [26]. Given the unknown epidemiology in Madagascar, MAT 209interpretation regarding serovars should be made with caution. .OK.IT’S FOR THIS REASON THAT INFECTIOUS SEROGROUPS ONLY CAN BE IDENTIFIED AND FOR THIS REASON TOO, IT IS GENERALLY BETTER TO USE MORE THAN ONE SEROVAR FOR A GIVEN SEROGROUP, ESPECIALLY ICTEROHAEMORRHAGIAE, THE MOST PROMINENT IN HUMANS. THE UNIC STRAIN USED IN THIS STUDY IS WELL KNOWN TO GIVE POOR AGGLUTINATING LEVEL, COMPARED TO OTHER ICTERO STRAINS AND TO THE CLOSE SEROVAR COPENHAGENI.:137No sample showed a positive reaction against serovar L. Canicola (L. Canicola) or L. Icterohaemorrhagiae (L. Icterohaemorrhagiae).
Thank you very much. We will consider this for subsequent studies in which the MAT will be applied. We used this Ictero-strain, because we received it from our German reference laboratory many years ago when we started working with Leptospirosis.
8. 214 To verify the diagnosis “leptospirosis” with certainty, a paired serum sample collected three weeks apart is 215 required. A fourfold or greater rise in antibody titers between the two sera confirms the diagnosis [25].THIS MAT INTERPRETATION IS TRUE FOR ACUTE CASES OF LEPTOSPIROSIS AS DIAGNOSED IN HUMANS AND CANINE WHICH EXHIBIT INDIVIDUAL DISEASE BUT NOT IN CATTLE. LEPTOSPIROSIS IN CATTLE IS NOT A DISEASE OF ONE ANIMAL. IT’S AN INFECTION SPREADING IN THE WHOLE HERD INDUCING SELDOM ACUTE LEPTOSPIROSIS EXPRESSED AS THE MILKING DROP SYNDROM, BUT MOSTLY ABORTIONS AND REPRODUCTIVE FAILURE WHICH ARE NOT ACUTE. THEREFORE HIGH TITERS CAN BE SHOWN BY THE NEIGHBOUR OF THE ABORTED COW WHEREAS THE TITERS OF THE ABORTED COW ITSELF YET DROPPED DOWN, BECAUSE AS INDICATED: 63The infection with serovar L. Hardjo, for which cattle are known to be a maintenance host, is often sublinical. .. 186Transmission of serovar L. Hardjo is possible through placental infection or direct contact with urine or fetal membranes from infected cattle [2]. In maintenance hosts, infection is often 187subclinical or the clinical signs are very mild.
As this interpretation was not correct, we deleted the appropriate parts in the discussion.
9. 262 We also thank Ivonne Stamm from IDEXX Laboratories, Ludwigsburg, Germany for her technical advice concerning the MAT.
WHY IDEXX?? THIS COMPANY IS NOT SPECIALIZED IN MAT, BUT IN OTHER METHODS AS ELISA....IT WOULD HAVE BE MORE JUSTIFIED TO DISCUSS WITH ACADEMIC OR REFERENCE VET CENTERS.ONSET OF HUMAN LEPTOSPIROSIS IS STRONGLY DIFFERENT FROM HERD LEPTOSPIROSIS? EVEN IF IMMUNOLOGICAL EVENTS ARE SIMILAR;
We worked together with IDEXX some years ago and they showed us how to culture leptospires correctly. As this works excellent in our lab we also asked them for the laboratory skills, we needed to establish the MAT in our lab.
Sincerely yours,
Theresa Schafbauer
Prof. Dr. Reinhard K. Straubinger
Reviewer 2 Report
This study reports seroprevalence estimates for Leptospira serovars obtained from slaughterhouses from five region sin Madagsacar. I have only minor comments at this point that I would like the authors to address.
Line 139: Does this refer to the seroprevalence of leptospires (positive to any serovar) among regions or an individual serovar comparison among regions? I’m assuming it was the former and it would be good to reinforce in line 113 that differences in “Leptospira” seroprevalence were analyzed.
On the same subject, if individual serovar comparisons by region were not done, I think it would be of great value to perform these analyses and report the findings in the text and/or perhaps a supplementary table with the individual serovar frequencies by region. I’m just curious to know whether there were regional seroprevalence differences at the serovar level.
Line 177: What MAT cutoff was used in these studies? It’d be good to add them in the text if different to 1:100.
Line 178: Same, please add the MAT titer cutoff used in the study cited for comparison.
Line 183: Do you know whether Hardjo interrogans or borgpetersenii is present in Madagascar or both? These two genospecies are likely to cross react in the MAT test. So it may be possible that Hardjo borgpetersenii antibodies were detected, no hardjo interrogans. Please clarify and address this issue in the discussion, if relevant.
Line 211: Please add the reference of the OIE statement about the use of 1:100 titer for a positive sample.
Lines 234-235: It is not clear to me what you mean with central and northern part in total? Do you mean prevalence estimates for northern part and central parts or just a seroprevalence estimate for Madagascar?
Author Response
Cover letter Munich, June 3rd, 2019
Dear Editors, dear Referees,
thank you for the quick processing of our manuscript “Seroprevalence of Leptospira spp. Infections in Cattle from Central and Northern Madagascar” and for your constructive comments. We provide our response and explanations to your comments in red fonts.
Concerning the report of the reviewer #2:
1. Line 139: Does this refer to the seroprevalence of leptospires (positive to any serovar) among regions or an individual serovar comparison among regions? I’m assuming it was the former and it would be good to reinforce in line 113 that differences in “Leptospira” seroprevalence were analyzed.
This refers to the Leptospira spp. seroprevalence among regions. We added “Leptospira spp.” seroprevalence in lines 116 and 143.
2. On the same subject, if individual serovar comparisons by region were not done, I think it would be of great value to perform these analyses and report the findings in the text and/or perhaps a supplementary table with the individual serovar frequencies by region. I’m just curious to know whether there were regional seroprevalence differences at the serovar level.
We calculated the seroprevalences of each serovar for every sampling region (line 117). There was no statistical significant difference. We included the information in the text “Moreover, there were no statistical significant differences between the seroprevalences of each serovar among the five different sampling regions (Table S1)” (lines 144/145) and created a supplementary table (Table S1).
3. Line 177: What MAT cutoff was used in these studies? It’d be good to add them in the text if different to 1:100.
They used a MAT titer cut-off of 1:48. We added this to the manuscript (lines 198/199).
4. Line 178: Same, please add the MAT titer cutoff used in the study cited for comparison.
They used a MAT titer cut-off of 1:50. We added this to our manuscript (lines 198/199).
5. Line 183: Do you know whether Hardjo interrogans or borgpetersenii is present in Madagascar or both? These two genospecies are likely to cross react in the MAT test. So it may be possible that Hardjo borgpetersenii antibodies were detected, no hardjo interrogans. Please clarify and address this issue in the discussion, if relevant.
Currently there is no information, which one of the two genospecies is present in Madagascar. We added this point to our discussion (lines 208-210).
“Currently information about the genospecies L. interrogans and L. borgpetersenii on Madagascar is missing. Thus, as these genospecies are likely to cross-react in MAT, it may be possible that the detected antibodies are against L. borgpetersenii serovar Hardjo but not L. interrogans serovar Hardjo.”
6. Line 211: Please add the reference of the OIE statement about the use of 1:100 titer for a positive sample.
We inserted the OIE reference into our text.
7. Lines 234-235: It is not clear to me what you mean with central and northern part in total? Do you mean prevalence estimates for northern part and central parts or just a seroprevalence estimate for Madagascar?
We refer to the prevalence estimate for northern and central part of Madagascar. We changed this sentence accordingly to make it more clearly (lines 264/265).
“Therefore, we did not calculate the seroprevalence for each sampling region but for the central and northern part of Madagascar in total.”
Sincerely yours,
Theresa Schafbauer
Prof. Dr. Reinhard K. Straubinger
Reviewer 3 Report
The study presented by the authors with the objective of identifying the prevalence of leptospira in cattle as a risk to public health, in 5 regions of Madagascar, in my opinion is adequate.
The introduction supports the study, the methodology is clear, the results are well presented, and the discussion is made according to the results and seeks to justify them, also referring to the limitations.
However, considering the data, if they could have explored a little more the factors that could be associated to the prevalence found, although in the discussion they mentioned possible causes, the data does not confirm it.
The conclusion could be a little more ambitious, in the sense that the future studies identify beyond the agents also possible causes. In addition to suggesting some public health measures to mitigate the problem.
Author Response
Cover letter Munich, June 3rd, 2019
Dear Editors, dear Referees,
thank you for the quick processing of our manuscript “Seroprevalence of Leptospira spp. Infections in Cattle from Central and Northern Madagascar” and for your constructive comments. We provide our response and explanations to your comments in red fonts.
Concerning the report of the reviewer #3:
1. However, considering the data, if they could have explored a little more the factors that could be associated to the prevalence found, although in the discussion they mentioned possible causes, the data does not confirm it.
We included possible factors for the detected prevalence in our discussion (lines 179-183).
“Factors associated with such a high seroprevalence in cattle may be due to the contact with other animals like domesticated or wild pigs, sheep, rodents or other small mammals, which are infected with leptospires and transmit them via urine. Another important source of infection could be the common usage of water sources. The transmission of leptospires via venereal spread is also a factor that should not be underestimated.”
2. The conclusion could be a little more ambitious, in the sense that the future studies identify beyond the agents also possible causes. In addition to suggesting some public health measures to mitigate the problem.
We extended our conclusion and added the points you mentioned (lines 286-290).
“Beyond the detection of leptospires it is also advised to search for possible causes of Leptospirosis on Madagascar. Moreover, humans on Madagascar should be informed adequately about this zoonotic disease and how they can prevent from being infected. For example by avoiding contact to rodents or urine of cattle or the use of protective gloves during work with cattle.”
Sincerely yours,
Theresa Schafbauer
Prof. Dr. Reinhard K. Straubinger